# Symplectic Recurrent Neural Networks

**Zhengdao Chen**[a,c], **Jianyu Zhang**[b,c],
**Martin Arjovsky**[a], **Léon Bottou**[c,a]

[a] New York University, New York, USA
[b] Tianjin University, Tianjin, China
[c] Facebook AI Research, New York, USA

## Abstract

We propose Symplectic Recurrent Neural Networks (SRNNs) as learning algorithms that capture the dynamics of physical systems from observed trajectories. An SRNN models the Hamiltonian function of the system by a neural network and furthermore leverages symplectic integration, multiple-step training and initial state optimization to address the challenging numerical issues associated with Hamiltonian systems. We show SRNNs succeed reliably on complex and noisy Hamiltonian systems. We also show how to augment the SRNN integration scheme in order to handle stiff dynamical systems such as bouncing billiards.

## 1 Introduction

Can machines learn physical laws from data? A recent paper (Greydanus et al., 2019), Hamiltonian Neural Networks (HNN), proposes to do so representing the Hamiltonian function $H(q, p)$ as a multilayer neural network. The partial derivatives of this network are then trained to match the time derivatives $\dot{p}$ and $\dot{q}$ observed along the trajectories in state space.

The ordinary differential equations (ODEs) that express Hamiltonian dynamics are famous for both their mathematical elegance and their challenges to numerical integration techniques. Except maybe for the simplest Hamiltonian systems, discretization errors and measurement noise lead to quickly diverging trajectories. In other words, Hamiltonian systems can often be *stiff*, a concept that usually refers to differential equations where we have to take very small time-steps of integration so that the numerical solution remain stable (Lambert, 1991). A plethora of numerical integration methods, *symplectic integrators*, have been developed to respect the conserved quantities in Hamiltonian systems, thereby usually being more stable and structure-preserving than non-symplectic ones (Hairer et al., 2002). For example, the simplest symplectic integrator is the well-known leapfrog method, also known as the Stömer-Verlet integrator (Leimkuhler and Reich, 2005). However, even the best integrators remain severely challenged by phenomena as intuitive as a mechanical rebound or a slingshot effect, which are more severe forms of stiffness. Such numerical issues are almost doomed to conflict with the inherently approximate nature of a learning algorithm.

In the first part of this paper, we propose *Symplectic Recurrent Neural Networks* (SRNNs), where (*i*) the partial derivatives of the neural-network-parametrized Hamiltonian are integrated with the leapfrog integrator and where (*ii*) the loss is back-propagated through the ODE integration over multiple time steps. We find that in the presence of observation noise, SRNN are far more usable than HNNs. Further improvements are achieved by simultaneously optimizing the initial state and the Hamiltionian network, presenting an interesting contrast to previous literature on the hardness of general initial state optimization (Peifer and Timmer, 2007). The optimization can be motivated from a maximum likelihood estimation perspective, and we provide heuristic arguments for why the initial state optimization is likely convex given the symplecticness of the system. Furthermore, experiments in the three-body problem show that the SRNN-trained Hamiltonian compensates for discretization errors and can even outperform numerically solving the ODE using the true Hamiltonian and the same time-step size. This could be of particular interest to researchers who study the application of machine learning to numerically solving differential equations.

The second part of this paper focuses on *perfect rebound* as an example of the more severe form of stiffness. When a point mass rebounds without loss of energy on a perfectly rigid

obstacle, the motion of the point mass is changed in ways that can be interpreted as an infinite force applied during an infinitesimal time. The precise timing of this event affects the trajectory of the point mass in ways that essentially make it impossible to merely simulate the Hamiltonian system on a predefined grid of time points. In order to address such events in learning, we augment the leapfrog integrator used in our SRNN with an additional trainable operator that models the rebound events and relates their occurrence to visual hints. Training such an augmented SRNN on observed trajectories not only learns the point mass dynamics but also learns the visual appearance of the obstacles.

## 2 Related work

**Learning physics with neural networks** A popular category of methods attempts to replicate the intuitive ways in which humans perceive simple physical interactions, identifying objects and learning how they relate to each other (Battaglia et al., 2016; Chang et al., 2016). Since such methods cannot be used for more general physical systems, another category of methods seeks to learn which differential equations govern the evolution of a physical system on the basis of observed trajectories. Brunton et al. (2016) assemble a small number of predefined primitives in order to find an algebraically simple solution. Lutter et al. (2019) use a neural network to model the Lagrangian function of a robotic system. Most closely related to ours, Greydanus et al. (2019) use a neural network to learn the Hamiltonian of the dynamical system in such a way that its partial derivatives match the time derivatives of the position and momentum variables, which are both assumed to be observed. Although the authors show success on a simple pendulum system, this approach does not perform well on a more complex system such as a three-body problem.

**ODE-based learning and recurrent neural networks (RNNs)** To learn an ODE that underlies some observed time series data, Chen et al. (2018a) proposes to solve a neural-network-parameterized ODE numerically and minimize the distance between the generated time series with the observed data. To save memory, they propose to use the adjoint ODE instead of back-propagating through the ODE solver. Using stability analysis of ODEs, Chang et al. (2019) propose the AntisymmetricRNN with better trainability. Niu et al. (2019) establish a correspondence between RNNs and ODEs, and propose an RNN architecture inspired by a universal quantum computation scheme.

*Summary of our main contributions:* In this paper, we propose SRNN, which

- learns Hamiltonian dynamics directly from position and momentum time series
- performs well on noisy and complex systems such as a spring-chain system and a three-body system, and is compatible with initial state optimzation
- is augmented to handle perfect rebound, an example of very stiff Hamiltonian dynamics

## 3 Framework

### 3.1 Hamiltonian systems

A Hamiltonian system of dimension $d$ is described by two vectors $p, q \in \mathbb{R}^d$. Typically, they correspond to the momentum and position variables, respectively. The evolution of the system is determined by the Hamiltonian function $H : (p, q, t) \in \mathbb{R}^{2d+1} \mapsto H(p, q, t) \in \mathbb{R}$ through a system of ordinary differential equations called *Hamilton's equations*,

$$\dot{p} = -\frac{\partial H}{\partial q} \ , \qquad \dot{q} = +\frac{\partial H}{\partial p} \ , \tag{1}$$

where we use the dot notation to compactly represent derivatives with respect to the time variable $t$. We are focusing in this work on Hamiltonians that are *conservative*,[1] that is, they do not depend on the time variable $t$, and *separable*,[2] that is, they can be written as a

---

[1] In our opinion, extending this work to non-conservative systems should not be done by adding a time dependency in the Hamiltonian, but by adding additional dissipation or intervention operators in the numerical integration schema, as illustrated in section 6.

[2] Extending this work to non-separable Hamiltonians can be achieved by rewriting the numerical integration schema using an extended phase space (Tao, 2016).

sum $H(p, q) = K(p) + V(q)$. In this case, (1) becomes

$$\dot{p} = -V'(q), \ \dot{q} = K'(p) \tag{2}$$

With a proper choice of the $p$ and $q$ variables, the evolution of essentially all physical systems can be described with the Hamiltonian framework. In other words, Hamilton's equations restrict the vast space of dynamical systems to the considerably smaller space of dynamical systems that are physically plausible.

Therefore, instead of modeling the dynamics of a physical system with a neural network $f_\theta(p, q)$ whose outputs are interpreted as estimates of the time derivatives $\dot{p}$ and $\dot{q}$, we can also use a neural network $H_\theta(p, q) = K_{\theta_1}(p) + V_{\theta_2}(q)$ with $\theta = [\theta_1, \theta_2]$, whose partial derivatives $-V'_{\theta_2}(q)$ and $K'_{\theta_1}(p)$ are interpreted as the time derivatives $\dot{p}$ and $\dot{q}$. We refer to the former as ODE neural networks (O-NET) and the latter approach as Hamiltonian neural networks (H-NET). In order to define a complete learning system, we need to explain how to determine the parameter $\theta$ of the neural networks on the basis of observed discrete trajectories. For instance, Greydanus et al. (2019) trains H-NET in a fully supervised manner using the observed tuples $(p, q, \dot{p}, \dot{q})$.

### 3.2 FROM ODEs TO DISCRETE TRAJECTORIES

A numerical integrator (or ODE solver) approximates the true solution of an ODE of the form $\dot{z} = f(z, t)$ at discrete time steps $t_0, t_1 \ldots t_T$. For instance, the simplest integrator, Euler's integrator, starts from the initial state $z_0$ at time $t_0$ and estimates the function $z(t)$ at uniformly spaced time points $t_n = t_0 + n \Delta t$ with the recursive expression

$$z_{n+1} = z_n + \Delta t \, f(z_n, t_n) \tag{3}$$

In stiff ODE systems, however, using Euler's method could easily lead to unstable solutions unless the time-step is chosen to be very small (Lambert, 1991). The development of efficient and accurate numerical integrators is the object of considerable research (Hairer et al., 2008; Hairer and Wanner, 2013). Symplectic integrators[3] are particularly attractive for the integration of Hamilton's equations (Leimkuhler and Reich, 2005). They are able to preserve quadratic invariants, and therefore usually have desired stability properties as well as being structure-preserving (McLachlan et al., 2004), even for certain non-Hamiltonian systems (Chen et al., 2018b). A simple and widely-used symplectic integrator is the leapfrog integrator. When the Hamiltonian is conservative and separable (2), it computes successive estimates $(p_n, q_n)$ with

$$
\begin{aligned}
p_{n+1/2} &= p_n - \tfrac{1}{2} \Delta t \, V'(q_n) \\
q_{n+1} &= q_n + \Delta t \, K'(p_{n+1/2}) \\
p_{n+1} &= p_{n+1/2} - \tfrac{1}{2} \Delta t V'(q_{n+1})
\end{aligned}
\tag{4}
$$

Repeatedly executing update equations (4) is called the *leapfrog algorithm*, which is as computationally efficient as Euler's method yet considerably more accurate when the ODE belongs to a Hamiltonian system (Leimkuhler and Reich, 2005).

### 3.3 LEARNING ODEs FROM DISCRETE TRAJECTORIES

Following Chen et al. (2018a), let the right hand side of the ODE be a parametric function $f_\theta(z, t)$ and let $z_0 \ldots z_T$ be an observed trajectory measured at uniformly spaced time points $t_0 \ldots t_T$. We can estimate the parameter $\theta$ that best represents the dynamics of the observed trajectory by minimizing the mean squared error $\sum_{i=1}^{T} \|z_i - \hat{z}_i(\theta)\|_2$ between the observed trajectory $\{z_i\}_{i=0}^{T}$ and the trajectory $\{\hat{z}_i(\theta)\}_{i=0}^{T}$ generated with our integrator of choice,

$$\{\hat{z}_i(\theta)\}_{i=0}^{T} = Integrator(z_0, f_\theta, \{t_i\}_{i=0}^{T}) \ .$$

For instance, this minimization can be achieved using stochastic gradient descent after back-propagating through the steps of our numerical integration algorithm of choice and then through each call to the functions $f_\theta$. This can be done when $f_\theta(z)$ is a neural network (O-NET), or is the concatenation $[-V'_{\theta_2}(q), K'_{\theta_1}(p)]$ of the partial derivatives of an H-NET

---

[3]An integrator is symplectic if applied to Hamiltonian systems, its flow maps are symplectic for short enough time-steps. For details, see Hairer et al. (2002) and Leimkuhler and Reich (2005).

$H_\theta(p,q) = K_{\theta_1}(p) + V_{\theta_2}(q)$, where the partial derivatives can be expressed using the same parameters $\theta$ as the Hamiltonian $H_\theta(p,q)$, for instance using automatic differentiation. We can then predict trajectories at testing time using the trained $f_{\theta*}$ and initial state $z_0^{\text{test}}$,

$$\{\hat{z}_i^{\text{test}}\}_{i=0}^{T^{\text{test}}} = Integrator(z_0^{\text{test}}, f_{\theta*}, \{t_i\}_{i=0}^{T^{\text{test}}}) \ .$$

Note that neither the integrator, nor the number of steps, nor the step size, need to be the same at training and testing.

### 3.4 SYMPLECTIC RECURRENT NEURAL NETWORK

This framework provides a number of nearly orthogonal design options for the construction of algorithms that model dynamical systems using trajectories:

- The time derivative model could be an O-NET or H-NET.
- The training integrator can be any explicit integrators. In our experiments, we only focus on Euler's integrator and the leapfrog integrator.
- The training trajectories can consist of a single step, $T=1$, or multiple steps, $T>1$. We refer to the first case as *single-step* and the second case as *multi-step* or *recurrent* training, because back-propagating through multiple steps of the training integrator is comparable to back-propagating through time in recurrent networks.
- The testing integrator can also be chosen freely and can use a different time-step size as it does not involve back-propagation.

In order to save space while describing the possibly different integrators used for training and testing, we use the labels "E-E", "E-L", and "L-L", where the first letter tells which integrator was used for training —"E" for Euler and "L" for leapfrog— and the second letter indicates which integrator was used as testing time. For instance, with our terminology, the HNN model of Greydanus et al. (2019) is a "single-step E-E H-NET" with the additional subtlety that they supervise the training with actual derivatives instead of relying on finite differences between successive steps of the observed trajectories.

A *Symplectic Recurrent Neural Network* (SRNN) is a recurrent H-NET that relies on a symplectic integrator for both training and testing, such as, for instance, a "recurrent L-L H-NET". As shown in the rest of this paper, SRNNs are far more usable and robust than the alternatives, especially when the Hamiltonian gets complex and the data gets noisy. We believe that SRNNs may also have other potential benefits: because leapfrog preserves volumes in the state space (Hairer et al., 2002), we conjecture that vanishing and exploding gradients' issues in backpropagating through entire state sequences are ameliorated (Arjovsky et al., 2015). Finally, because the leapfrog integrator is reversible in time, there is no need to store states during the forward pass as they can be recomputed exactly during the backward pass. We leave studying these other computational and optimization benefits as a topic of future work.

## 4 SRNN CAN LEARN COMPLEX AND NOISY HAMILTONIAN DYNAMICS

As an example of a complex Hamiltonian system, we first present experiments performed on the spring-chain system: a chain of 20 masses with neighbors connected via springs. Each of the two masses on the ends are connected to fixed ground via another spring. The chain can be assumed to lay horizontally and the masses move vertically but no gravity is assumed. The 20 masses and the 21 spring constants are chosen randomly and independently. The training data consist of 1000 trajectories of the same chain, each of which starts from a random initial state of positions and momenta of the masses and is 10-time-step long (including the initial state). We thus take $T = 9$ when performing recurrent training. When performing single-step training, each training trajectory of length 10 is instead considered as 9 consecutive trajectories of length 2. In this way, 1000 sample trajectories of length 10 ($T$=9) are turned into 9000 sample trajectories of length 2 ($T$=1), allowing for a fair comparison between single-step training and recurrent training. During testing, the trained model is given 32 random initial states in order to predict 32 trajectories of length 100. Detailed experiment setups and model architectures are provided in Appendix A.1, and a PyTorch implementation can be found at https://github.com/zhengdao-chen/SRNN.git.

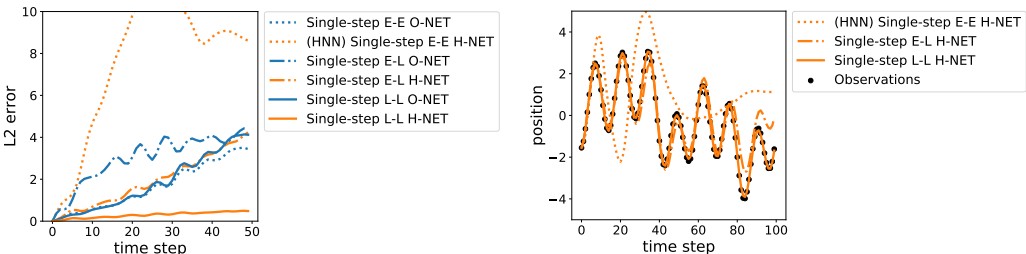

Figure 1: Testing results in the noiseless case by single-step methods. Left: Prediction error of each method over time, measured by the L2 distance between the true and predicted positions of the 20 masses. Right: Each curve represents the position of one of the masses (number 5) as a function of time predicted by the three single-step-trained H-NET models. Plots of the other masses' positions are provided in Appendix D.1.

### 4.1 Going symplectic - rescuing HNN with the leapfrog integrator

First, we consider the noiseless case, where the training data consist of exact values of the positions ($q$) and momenta ($p$) of the masses on the chain at each discrete time point. As shown in figure 1, the prediction of a single-step E-E H-NET deviates from the ground truth quickly and is unable to capture the periodic motion. By comparison, a single-step E-E O-NET yields predictions that is qualitatively reasonable. This shows that using Hamiltonian models without paying attention to the integration scheme may not be a good idea.

We then replace Euler's integrator used during testing by a leapfrog integrator, yielding a Single-step E-L H-NET. Figure 1 shows that this helps the H-NET produce predictions that remain stable and periodic over a longer period of time. Since the training process remains the same, this implies that part of the instability and degeneration of H-NET's predictions comes from the nature of Euler's integrator rather than the lack of proper training.

In contrast, using a leapfrog integrator for both training and testing substantially improve the performance, as also shown again in figure 1. This improvement shows the importance of consistency between the integrators used in training and predicting modes. This can be understood with the concept of *modified equations* (Hairer, 1994): when we use a numerical integrator to solve an ODE, the numerical solution usually does not strictly follow the original equation due to discretization, but can be regarded as a solution to a modified version of the original equation that depends on the integrator and the time-step size. Therefore, training and testing with the same numerical integrator and time-step size could allow the system to learn a modified Hamiltonian that corrects some of the errors caused by the discretization scheme.

### 4.2 Going recurrent - using multi-step training when noise is present

Since noise is prevalent in real-world observations, we also test our models on noisy trajectories. Independent and identically distributed Gaussian noise is added to both the position and the momentum variables at each time step. Applying the single-step methods described above yield considerably worse predictions, as shown in Figure 2 (left).

This phenomenon can be controlled by training on multiple steps, effectively arriving at a type of recurrent neural network: if noise is added independently at each time-step, then having data from multiple consecutive time steps may allow us to discern the actual noiseless trajectory, analogous to performing linear regression on multiple (more than 2) noisy data points. As we see in Figure 2 (left), recurrent training consistently improves the predictions except for E-E H-NET. The best performing model is the SRNN (recurrent L-L H-Net) which improves substantially over the single-step L-L H-NET. Interestingly, the recurrent E-E H-NET does not improve over the single-step E-E H-NET, which means that recurrent training does not help if one uses a naïve integrator.

### 4.3 Initial state optimization (ISO)

However, one issue remains to be addressed: in the framework that we have adopted so far, the initial states $p_0$ and $q_0$ are treated as the actual initial states from which the system

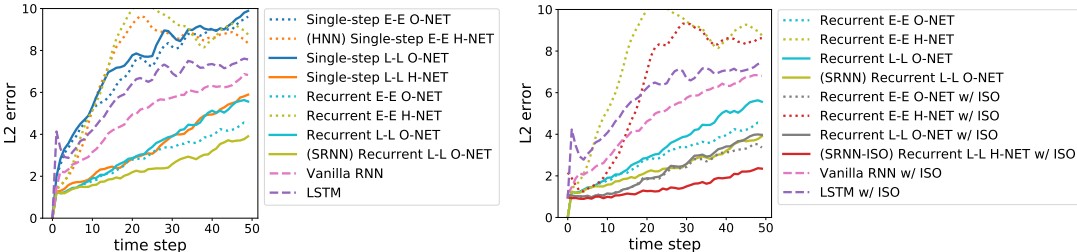

Figure 2: Prediction error of all methods in the noisy case measured by L2 distance, presented in two plots due to the large number of methods. Included in the left plot are the single-step-trained methods, recurrently trained methods, vanilla RNN and LSTM. Included in the right plot are the (same) recurrently trained methods, the recurrently trained methods with initial state optimization (ISO), as well as vanilla RNN and LSTM with ISO.

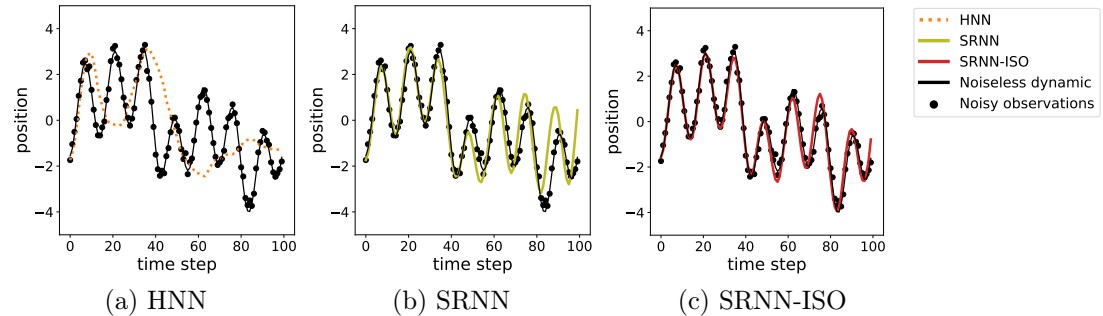

Figure 3: Predictions made by three methods in the noisy case. The Y-axis corresponds to the position of one of the masses (number 5) on the chain.

begins to evolve despite the added noise in observation. With noise added to the observation of $p_0$ and $q_0$, our dynamical models will start from these noisy states and remain biased as we advance in time in both the training and the testing mode.

To mitigate this issue, we propose to introduce two new parameter vectors for each sample, $\hat{p}_0$ and $\hat{q}_0$, interpreted as our estimate of the actual initial states, and we let our dynamical models evolve starting from them instead of the observed $p_0$ and $q_0$. Treating $\hat{p}_0$ and $\hat{q}_0$ as parameters, we can optimize them based on the loss function while fixing the model's parameters, a process that we call *initial state optimization* (ISO). When the model is good enough, we hope that this will guide us towards the true initial states without observation noise. In Appendix B, we motivate the use of ISO from the perspective of maximum likelihood inference. In actual training, we first train the neural network parameters for 100 epochs as usual, and starting from the 101st, after every epoch we perform ISO with the L-BFGS-B algorithm (Zhu et al., 1997) on the $\hat{p}_0$ and $\hat{q}_0$ parameters for every training trajectory. At testing time, the model is given the noisy values of $p$ and $q$ for the first 10 time steps and must complete the trajectory for the next 200 steps. These 10 initial time steps allow us to perform the same L-BFGS-B optimization to determine the initial state $\hat{p}_0$ before advancing in time to predict the entire trajectory.

As seen in Figure 2 (right), SRNN-ISO (i.e. SRNN equipped with ISO) clearly yields the best prediction among all the methods. Figure 3 shows the predictions of HNN, SRNN and SRNN-ISO on one test sample, and we clearly see the qualitative improvements thanks to recurrent training and ISO. O-NET also benefits from ISO while vanilla RNN and LSTM do not seem to, likely because the initial state optimization only works when we already have a reasonable model of the system. In Appendix C, we give a heuristic argument for the convexity of ISO, which helps to explain the success of using L-BFGS-B for ISO.

In summary, we have proposed three extensions to learning complex and noisy dynamics with H-NET and demonstrated the improvements they lead to: a) using the leapfrog integrator instead of Euler's integrator; b) using recurrent instead of single-step training; and c)

Table 1: Testing results of predicting the dynamics of the spring-chain system by methods based on fixed $p_0$, $q_0$ (i.e., not optimizing $p_0$, $q_0$ as parameters). The error is defined as the discrepancy between the (noisy) ground truth and the predictions at each time step averaged over the first 200 time steps, where the discrepancy is measured by the L2 distance between the true and predicted positions of the 20 masses in the chain, both of which considered as 20-dimensional vectors. The mean and standard deviation are computed based on 32 testing samples, each starting from a random configuration of the chain.

| | Model | Integrator (tr) | Integrator (te) | Error mean | Error std |
|---|---|---|---|---|---|
| single-step | O-NET | Euler | Euler | 6.93 | 1.22 |
| | | Euler | Leapfrog | 5.87 | 1.04 |
| | | Leapfrog | Leapfrog | 7.28 | 1.48 |
| | H-NET | Euler | Euler | 7.24 | 0.64 |
| | | Euler | Leapfrog | 3.32 | 0.89 |
| | | Leapfrog | Leapfrog | 3.36 | 0.67 |
| recurrent | O-NET | Euler | Euler | 2.88 | 0.45 |
| | | Euler | Leapfrog | 4.12 | 0.41 |
| | | Leapfrog | Leapfrog | 3.34 | 0.86 |
| | H-NET | Euler | Euler | 7.58 | 0.63 |
| | | Euler | Leapfrog | 5.26 | 0.63 |
| | | Leapfrog | Leapfrog | **2.37** | **0.87** |
| | Vanilla RNN | N/A | N/A | 4.80 | 0.82 |
| | LSTM | N/A | N/A | 5.95 | 1.05 |

Table 2: Testing results of predicting the dynamics of the spring-chain system by methods that optimize on $p_0$ and $q_0$ starting from their observed (noisy) values using L-BFGS-B, as explained in the text. The definition of the errors is the same as in the above table.

| Model | Integrator (tr) | Integrator (te) | Error mean | Error std |
|---|---|---|---|---|
| O-NET | Euler | Euler | 2.13 | 0.37 |
| | Euler | Leapfrog | 3.59 | 0.50 |
| | Leapfrog | Leapfrog | 2.27 | 0.60 |
| H-NET | Euler | Euler | 6.26 | 0.60 |
| | Euler | Leapfrog | 3.00 | 0.63 |
| | Leapfrog | Leapfrog | **1.45** | **0.32** |
| Vanilla RNN | N/A | N/A | 4.72 | 0.94 |
| LSTM | N/A | N/A | 5.81 | 0.98 |

optimizing the initial states of each trajectory as parameters when data are noisy. Thorough comparisons of test errors are given in Tables 1 and 2, where we highlight that the SRNN (recurrent L-L H-NET) models achieve the lowest errors.

## 5   SRNN CAN LEARN THE DYNAMICS OF A THREE-BODY SYSTEM

Next, we test SRNN with the three-body system, which is a well-known example of a chaotic system, meaning that a small difference in the initial condition could lead to drastically different evolution trajectories, even without noise added. As a result, even when the exact equations are known, simulating it with different time-step sizes could also lead to qualitatively different solutions. Moreover, Greydanus et al. (2019) mentions that HNN does not outperform a baseline method using O-NET in learning the three-body system's evolution. Here, we test our SRNN together with other baselines on the noiseless three-body system with the same configurations as Greydanus et al. (2019). The detailed experimental setup and model architectures are provided in Appendix A.2.

As we see in Table 3, the best-performing model is SRNN and the second-best is the single-step L-L H-NET. Interestingly, and perhaps counter-intuitively, they even outperform the baseline method of simulating the correct equation with the same time-step size. How is this possible? In short, our explanation is that the error introduced by numerical discretization could be learned and therefore compensated for by the models we train. More concretely, once again using the concept of modified equations mentioned in Section 4.1, we argue that the ODE-based learning models, including both H-NET and O-NET models, could learn *not*

Table 3: Prediction error results for the three-body system with time-step $\Delta t = 1$. The last row corresponds to numerically solving the correct underlying equations using the leapfrog integrator with time-step $\Delta t = 1$. The other rows correspond to the different learning-based methods, same as in the spring-chain experiments.

|  | Model | Integrator (tr) | Integrator (te) | Error mean | Error std |
|---|---|---|---|---|---|
| single-step | O-NET | Euler | Euler | 0.65 | 0.16 |
|  |  | Euler | Leapfrog | 1.36 | 0.18 |
|  |  | Leapfrog | Leapfrog | 1.33 | 0.20 |
|  | H-NET | Euler | Euler | 1.64 | 0.25 |
|  |  | Euler | Leapfrog | 0.88 | 0.33 |
|  |  | Leapfrog | Leapfrog | 0.35 | 0.09 |
| recurrent | O-NET | Euler | Euler | 0.51 | 0.11 |
|  |  | Euler | Leapfrog | 1.27 | 0.18 |
|  |  | Leapfrog | Leapfrog | 0.49 | 0.10 |
|  | H-NET | Euler | Euler | 0.79 | 0.17 |
|  |  | Euler | Leapfrog | 1.76 | 0.62 |
|  |  | Leapfrog | Leapfrog | **0.26** | **0.07** |
| simulation | true eqns. | (no training) | Leapfrog | 0.47 | 0.18 |

the correct underlying equation, but rather the equation whose modified equation associated with our choice of numerical integrator and time-step size is the original equation. Hence, when the time-step size is large and the error of numerical discretization is not negligible, it is possible that the learned equation could yield better predictions than the correct one.

In addition, we also see that the recurrently trained models outperform the corresponding single-step-trained models. Plots of the predicted trajectories are provided in Appendix E.

## 6   Learning perfect rebound with an augmented SRNN

We focus in this section on the *perfect rebound* problem as a prototypical example of stiff ODE in a physical system. We consider a heavy billiard, subject to gravitational forces pointing downwards, and bouncing around a two-dimensional square domain delimited by impenetrable walls. Whenever it hits a wall, the billiard rebounds without loss of energy, by reversing the component of its momentum orthogonal to the wall surface. Microscopically, when the billiard hits the wall, the atomic structure deformation produces strong electro-magnetic forces that reverse the momentum during a very brief timescale. Simulating this microscopic phenomenon with a Hamiltonian ODE would not only be computationally expensive, but also require a detailed knowledge of the atomic structures of the billiard and the walls. The perfect rebound is a macroscopic approximation that treats the billiard as a point mass and the rebound as an event with zero duration infinite forces. Although this approximation is convenient for high-school level derivations, the singularity makes it hard to simulate using Hamiltonian dynamics.

We propose to approach this problem by augmenting each time step of a leapfrog-based SRNN with an additional operation that models a possible rebound event,

$$p_t^{post} \leftarrow p_t^{pre} - 2(p_t^{pre} \cdot n)n \ , \tag{5}$$

where $p_t^{pre}$ is the pre-rebound momentum vector and $p_t^{post}$ is the post-rebound momentum vector. When the vector $n$ is zero, this operation does not change the momentum in any way. When $n$ is a unit vector orthogonal to a wall, this operation computes the momentum reversal that is characteristic of a perfect rebound. Vectors $n$ of smaller length could also be used to model energy dissipation in manner that is reminiscent of the famous LSTM forget gate (Hochreiter and Schmidhuber, 1997).

Because the billiard trajectory depends on the exact timing of the rebound event, we also need a scalar $\alpha \in [0, 1]$ that precisely places the rebound event at time $t + \alpha\Delta t$ between the successive time steps $t$ and $t + \Delta t$. The augmented leapfrog schema then becomes

$$[p_{t+\alpha\Delta t}^{pre}, q_{t+\alpha\Delta t}^{pre}] \xleftarrow[\alpha\Delta t]{leapfrog} [p_t, q_t] \tag{6}$$

$$p_{t+\alpha\Delta t}^{post} = p_{t+\alpha\Delta t}^{pre} - 2(p_{t+\alpha\Delta t}^{pre} \cdot n)n \tag{7}$$

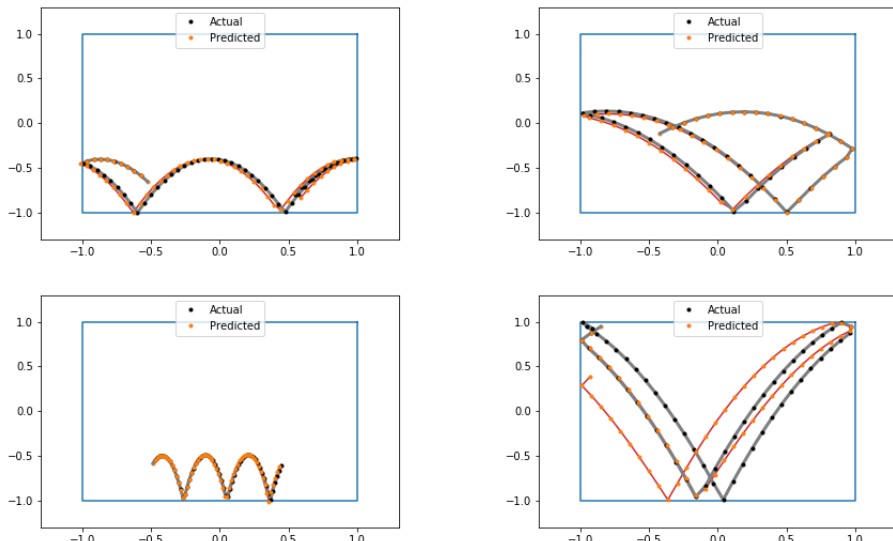

Figure 4: Actual versus predicted trajectories of the heavy billiard with perfect rebound. The predictions are obtained by an SRNN plus the rebound module described in section 6.

$$[p_{t+\Delta t}, q_{t+\Delta t}] \xleftarrow[(1-\alpha)\Delta t]{leapfrog} [p^{post}_{t+\alpha\Delta t}, q^{post}_{t+\alpha\Delta t}] \tag{8}$$

where equations (6) and (8) represent ordinary leapfrog updates (4) for time steps of respective durations $\alpha\Delta t$ and $(1-\alpha)\Delta t$. More precisely, we first compute a tentative position $\tilde{q}_{t+\Delta t}$ and momentum $\tilde{p}_{t+\Delta t}$ assuming no rebound,

$$[\tilde{p}_{t+\Delta t}, \tilde{q}_{t+\alpha\Delta t}] \xleftarrow[\Delta t]{leapfrog} [p_t, q_t] \ , \tag{9}$$

then compute both $n$ and $\alpha$ as parametric functions of the tentative position $\tilde{q}_{t+\Delta t}$ as well as the current position $q_t$, and finally apply the forward model (6–8). Note that the final state is equal to the tentative state when no rebound occurs, that is, when $n = 0$.

Directly modeling $n$ and $\alpha$ with a neural network taking $\tilde{q}_{t+\Delta t}$ as the input would be very inefficient because we would need to train with a lot of rebound events to precisely reveal the location of the walls. We chose instead to use *visual cues* in the form of a background image representing the walls. We model $n$ as the product of a direction vector $\bar{n}$ and a magnitude $\gamma \in [0, 1]$, and we want the latter to take value close to 1 when perfect rebound actually occurs between $t$ and $t + \Delta t$ and close to 0 otherwise. Both $\bar{n}$ and $\alpha$ are modeled as MLPs that take as input two 10x10 neighborhoods of the background image, centered at positions $q_t$ and $\tilde{q}_{t+\Delta t}$, respectively. In contrast, $\gamma$ is modeled as an MLP that takes as input a smaller 2x2 neighborhood centered at $\tilde{q}_{t+\Delta t}$ and is trained with an additional regularization term $\|\gamma\|_1$ in order to switch the rebound module off when it is not needed.

Training is achieved by back-propagating through the successive copies of the augmented leapfrog scheme, through the models of $\bar{n}$, $\alpha$, and $\gamma$, and also through the computation of the tentative $\tilde{p}_{t+\Delta t}$ and $\tilde{q}_{t+\Delta t}$. We use 5000 training trajectories of length 10 starting from a randomly-sampled initial positions and velocities. Similarly, we use 32 testing trajectories of length 60. Detailed exprimental setup is included in Appendix A.3. Figure 5 plots some predicted and actual testing trajectories. Appendix F compares these results with the inferior results obtained with several baseline methods, including SRNN without the rebound module, and SRNN with a rebound module that does not learn $\alpha$. One limitation of our method, however, results from the assumption that there is at most one rebound event per time step. Although this assumption fails when the billiard rebounds twice near a corner, as shown in the bottom right plot in Figure 5, our method still outperforms the baseline methods even in this case.

## 7 Conclusion

We propose the Symplectic Recurrent Neural Network, which learns the dynamics of Hamiltonian systems from data. Thanks to symplectic integration, multi-step training and initial state optimization, it outperforms previous methods in predicting the evolution of complex and noisy Hamiltonian systems, such as the spring-chain and the three-body systems. It can even outperform simulating with the exact equations, likely by learning to compensate for numerical discretization error. We further augment it to learn perfect rebound from data, opening up the possibility to handle stiff systems using ODE-based learning algorithms.

## Acknowledgments

The authors acknowledge stimulating discussions with Dan Roberts, Marylou Gabrié, Anna Klimovskaia, Yann Ollivier and Joan Bruna.

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

# A    EXPERIMENT SETUP

## A.1    THE SPRING-CHAIN EXPERIMENT

We set $\Delta t = 0.1$. The ground truth trajectories in both training and testing are simulated by the leapfrog integrator using $\Delta t' = 0.001$ and coarsened into time-grids of 0.1 with a factor of 100, since simulating with a much smaller time-step leads to much more accurate solution, which we will treat as the ground truth solution.

The O-NET that represents $f_\theta(p, q)$ is a one-hidden-layer MLP with 40 input units, 2048 hidden units and 40 output units. The H-NET that represents $H_\theta(p, q) = K_{\theta_1}(p) + V_{\theta_2}(q)$ consists of two one-hidden-layer MLPs, one for $K_{\theta_1}$ and the other for $V_{\theta_2}$. Each of the MLPs have 20 input units, 2048 hidden units and 1 output unit. The vanilla RNN and LSTM models also have hidden states of size 2048. Implemented in PyTorch, the models are trained over 1000 epochs with the Adam optimizer (Kingma and Ba, 2014) with initial learning rate 0.001 and using the *ReduceLROnPlateau* scheduler[4] with patience 15 and factor 0.7.

## A.2    THE THREE-BODY EXPERIMENT

The ground truth trajectories are simulated by SciPy's `solve_ivp` adaptive solver[5] with method RK45. We coarse-grain the simulated ground truth trajectories into time-steps of $\Delta t = 1$, so that the models developed in section 4 are numerically integrated with time-step $\Delta t = 1$ in both training and testing. We intentionally set the time-step to be relatively large, so that it becomes interesting to compare these models with a baseline method of simulating the true equations with time-step $\Delta t = 1$. In addition, the training data consist of 100 sample trajectories of length $10 \cdot \Delta t = 10$, which are then turned into 900 trajectories of length 2 and 600 trajectories of length 5, respectively for single-step and recurrent training, in the same way as for the spring-chain experiments above.

The O-NET that represents $f_\theta(p, q)$ is a three-hidden-layer MLP with 12 input units, 512 hidden units in each hidden layer and 12 output units. The H-NET that represents $H_\theta(p, q) = K_{\theta_1}(p) + V_{\theta_2}(q)$ consists of two three-hidden-layer MLPs, one for $K_{\theta_1}$ and the other for $V_{\theta_2}$. Each of the MLPs have 6 input units, 512 hidden units in each hidden layer and 1 output unit. The vanilla RNN and LSTM models also have hidden states of size 512. Implemented in PyTorch, the models are trained over 1000 epochs with the Adam optimizer with initial learning rate 0.0003 and using the *ReduceLROnPlateau* scheduler with patience 15 and factor 0.7.

## A.3    THE HEAVY BILLIARD EXPERIMENT

The full image has size 128x128 pixels. The thickness of the wall is 12 pixels on each of the four sides, which leaves the free space of size 104x104 pixels in the middle for the billiard to move within. The billiard has size 3x3 pixels.

The O-NET that represents $f_\theta(p, q)$ is a one-hidden-layer MLP with 4 input units, 32 hidden units and 4 output units. The H-NET that represents $H_\theta(p, q) = K_{\theta_1}(p) + V_{\theta_2}(q)$ consists of two one-hidden-layer MLPs, one for $K_{\theta_1}$ and the other for $V_{\theta_2}$. Each of the MLPs have 2 input units, 32 hidden units and 1 output unit. The vanilla RNN model also has hidden states of size 32. For the rebound module, $\bar{n}$ is computed as the normalized output of a two-hidden-layer MLP, with 200 input units, 128 units in the first hidden layer, 32 units in the second hidden layer and 2 output units. $\alpha$ is also computed using a two-hidden-layer MLP, sharing the first hidden layer units with the MLP for $\bar{n}$, and having 32 units in the second hidden layer and 1 output unit. $\gamma$ is computed by passing through sigmoid the output of a two-hidden-layer MLP, with 4 input units, 16 units in each hidden layer and 1 output unit. All of the activation functions are tanh except for the hidden-to-output activation in the MLP for $\alpha$, where ReLU is used. Implemented in PyTorch, the models are trained over 1500 epochs with the Adam optimizer with initial learning rate 0.005 and using the *ExponentialLR* scheduler[6] with decay factor 0.99 until the learning rate reaches 0.0001. We set $\Delta t = 0.1$, and use 5000 trajectories of length $10 \cdot \Delta t = 1$ as training data, and 32 trajectories of length $60 \cdot \Delta t = 6$ as testing data.

---

[4] https://pytorch.org/docs/stable/optim.html#torch.optim.lr_scheduler.ReduceLROnPlateau
[5] https://docs.scipy.org/doc/scipy/reference/generated/scipy.integrate.solve_ivp.html
[6] https://pytorch.org/docs/stable/optim.html#torch.optim.lr_scheduler.ExponentialLR

# B    THE MAXIMUM LIKELIHOOD ESTIMATION PERSPECTIVE

In the presence of noise, we can interpret the learning problem described in section 3.3 above from the perspective of maximum likelihood inference, which also provides justification for treating the initial states as trainable parameters. We define models as follow:

$$\hat{z}_i(\theta) = Integrator(\hat{z}_0 = z_0, f_\theta, \{t_i\}_{i=0}^T)$$

$$q(z_i; \theta) = \frac{1}{\sqrt{(2\pi)^d \sigma^{2d}}} e^{-\|z_i - \hat{z}_i(\theta)\|_2^2/(2\sigma^2)} \tag{10}$$

$$P(\{z_i\}_{i=1}^n | \theta) = \prod_{i=1}^n q(z_i; \theta) = \frac{1}{(\sqrt{2\pi\sigma^2})^{nd}} \prod_{i=1}^n e^{-\|z_i - \hat{z}_i(\theta)\|_2^2/(2\sigma^2)},$$

and $\mathcal{L}(\theta|\{z_i\}_{i=1}^n) = P(\{z_i\}_{i=1}^n|\theta)$ is the likelihood function given the time-series data $\{z_i\}_{i=1}^n$. Note that this model assumes independence between $z_i$ and $z_j$ for $i \neq j$ once $\theta$ is fixed.

If we are to perform maximum likelihood inference, we arrive at the following:

$$\max_\theta : \quad \log \mathcal{L}(\theta|\{z_i\}_{i=1}^n) = -\frac{nd}{2} \log(2\pi\sigma^2) - \frac{1}{2\sigma^2} \sum_{i=1}^n \|z_i - \hat{z}_i(\theta)\|_2^2, \tag{11}$$

which is equivalent to

$$\min_\theta \quad \sum_{i=1}^n \|z_i - \hat{z}_i(\theta)\|_2^2 \tag{12}$$

This provides a motivation for using the $L^2$ loss, as we did in the experiments.

So far, we consider $\theta$ as the only parameter of the model defined by equations 10, and therefore the only argument of the likelihood function, while $\hat{z}_0$ is fixed to be the observed initial state $z_0$. As a generalization, we can consider a strictly larger family of models by allowing $z_0$ to vary as well. In this way, we treat both $\theta$ and $z_0$ as the parameters in the model and therefore arguments of the likelihood function that we optimize on. In other words, the model becomes

$$\hat{z}_i(\theta, \hat{z}_0) = Integrator(\hat{z}_0, f_\theta, \{t_i\}_{i=0}^T)$$

$$q(z_i; \theta, \hat{z}_0) = \frac{1}{\sqrt{(2\pi)^d \sigma^{2d}}} e^{-\|z_i - \hat{z}_i(\theta, \hat{z}_0)\|_2^2/(2\sigma^2)} \tag{13}$$

$$P(\{z_i\}_{i=1}^n | \theta, \hat{z}_0) = \prod_{i=1}^n q(z_i; \theta) = \frac{1}{(\sqrt{2\pi\sigma^2})^{nd}} \prod_{i=1}^n e^{-\|z_i - \hat{z}_i(\theta)\|_2^2/(2\sigma^2)},$$

and the optimization problem becomes

$$\min_{\theta, \hat{z}_0} \quad \sum_{i=1}^n \|z_i - \hat{z}_i(\theta, \hat{z}_0)\|_2^2, \tag{14}$$

which justifies optimizing over the initial states $p_0$, $q_0$ in addition to the neural network parameters $\theta$ as described in the previous section.

Such an interpretation is similar to approaches for parameter estimation in the literature of inverse problems and systems biology, though in those cases the parameters of interest appear directly in ODEs instead of via neural networks (Peifer and Timmer, 2007; Stapor et al., 2018). In particular, jointly optimizing the parameters in the model as well as the initial value is called the initial value approach. However, despite the success we demonstrate in section 4.2, two difficulties of this approach have been pointed out: 1) The optimization could converge to local minima; 2) The numerical solution of the ODE can be unstable (Peifer and Timmer, 2007). As explained in section 4.1, using HNN together with the leapfrog integrator mitigates the second issue. But what about the first issue? In particular, even if we assume that the optimization of the neural network can work "magically" well and do not suffer from bag local minima, what about optimizing the initial value $\hat{z}_0$?

# C    SYMPLECTICNESS AND INITIAL-STATE-OPTIMIZATION CONVEXITY

The success of optimizing on the initial state of the system in addition to the recurrent H-NET and O-NET models as described in section 4.2 raises the following question: If we

already have a relatively well-trained H-NET or O-NET, is the optimization on the initial values convex? We formalize the question below and provide a heuristic answer.

For simplicity, we restrict our attention to autonomous ODEs, which means that the function $f$ in $\frac{dz}{dt} = f(z)$ does not depend on $t$. Assuming existence and uniqueness of solutions, there exists a function $\phi_t$ that maps each initial state $\hat{z}_0$ to the state of the system after evolving from $\hat{z}_0$ for time $t$, $\phi_t(\hat{z}_0)$. This function is usually called the flow map. Flow maps have also been defined for numerical solutions of ODEs, by letting $\phi_t(\hat{z}_0) = Integrator(z_0, f, \{t_i\}_{i=0}^T)$ with $t_0 = 0$. We can extend this definition to all the trainable models we have considered, including the models based on O-NET and H-NET by defining $\phi_t(\hat{z}_0)$ to be the state of the system after letting the system evolve from initial state $\hat{z}_0$ for time $t$, for suitable choices of $t$. For example, for O-NET, we have $\phi_t(\hat{z}_0) = Integrator(z_0, f_\theta, \{t_i\}_{i=0}^T)$.

Suppose we impose an L2 loss on $\phi_t(\hat{z}_0)$, $e_t(\hat{z}_0) = \|\phi_t(\hat{z}_0) - z_t\|_2^2$, where $z_t$ corresponds to the observed data at time $t$. The question is, is $e_t(z)$ a (perhaps locally) convex function of $z$, for what functions and numerical integrators? To understand convexity, we compute the gradient and the Hessian as follow.

$$\frac{\partial}{\partial z} e_t(z) = 2(\phi_t(z) - z_t)^\intercal \cdot F_t(z) \tag{15}$$

$$\frac{\partial^2}{\partial z^2} e_t(z) = 2(\phi_t(z) - z_t) \cdot^{(3)} G_t(z) + F_t(z)^\intercal \cdot F_t(z), \tag{16}$$

where $F_t(z)$ is the Jacobian matrix of the flow map, defined as $F_t(z)_{ij} = \frac{\partial}{\partial z_j}(\phi_t(z)_i)$, and $G_t(z)$ is a third-order tensor contains the second order derivatives of the flow map, defined as $G_t(z)_{ijk} = \frac{\partial^2}{\partial z_i z_j}(\phi_t(z)_k)$. We use $\cdot^{(3)}$ to denote the dot product in the third dimension.

$F_t(z)^\intercal \cdot F_t(z)$ is symmetric positive semidefinite for any matrix $F_t(z)$. If $\phi_t$ corresponds to either the exact flow map of a Hamiltonian system or the flow of a symplectic integrator, such as the leapfrog integrator, applied to a Hamiltonian system, then $F_t(z)$ is a symplectic matrix, implying that $\det(F_t(z)) = 1$. Hence, $\det(F_t(z)^\intercal \cdot F_t(z)) = 1$, which further implies that $F_t(z)^\intercal \cdot F_t(z)$ is a positive definite matrix. Therefore, non-rigorously, when $\|\phi_t(z) - z_t\|_2$ is small and so the first term on the right hand side of equation 16 is negligible compared to the least eigenvalue of $F_t(z)^\intercal \cdot F_t(z)$, the entire Hessian matrix $\frac{\partial^2}{\partial z^2} e_t(z)$ is also positive definite, implying strong convexity of the optimization problem.

If $\phi_t$ is the exact flow map, then $\|\phi_t(z) - z_t\|$ being small means that noise in the data is small. If $\phi_t$ is the flow map of a learned model, then it means that we have a model close to the true underlying system in addition to not having too much noise in the data. Translating back to the learning problem, we see that, heuristically, when the model we use is close to symplectic, which is likely if the underlying system is a Hamiltonian system, and trained to be close enough to the true underlying system, and the noise in the data is small enough, then the optimization problem on the initial state is strongly convex.

# D   ADDITIONAL PLOTS OF THE SPRING-CHAIN EXPERIMENTS

## D.1   NOISELESS DATA (SECTION 4.1)

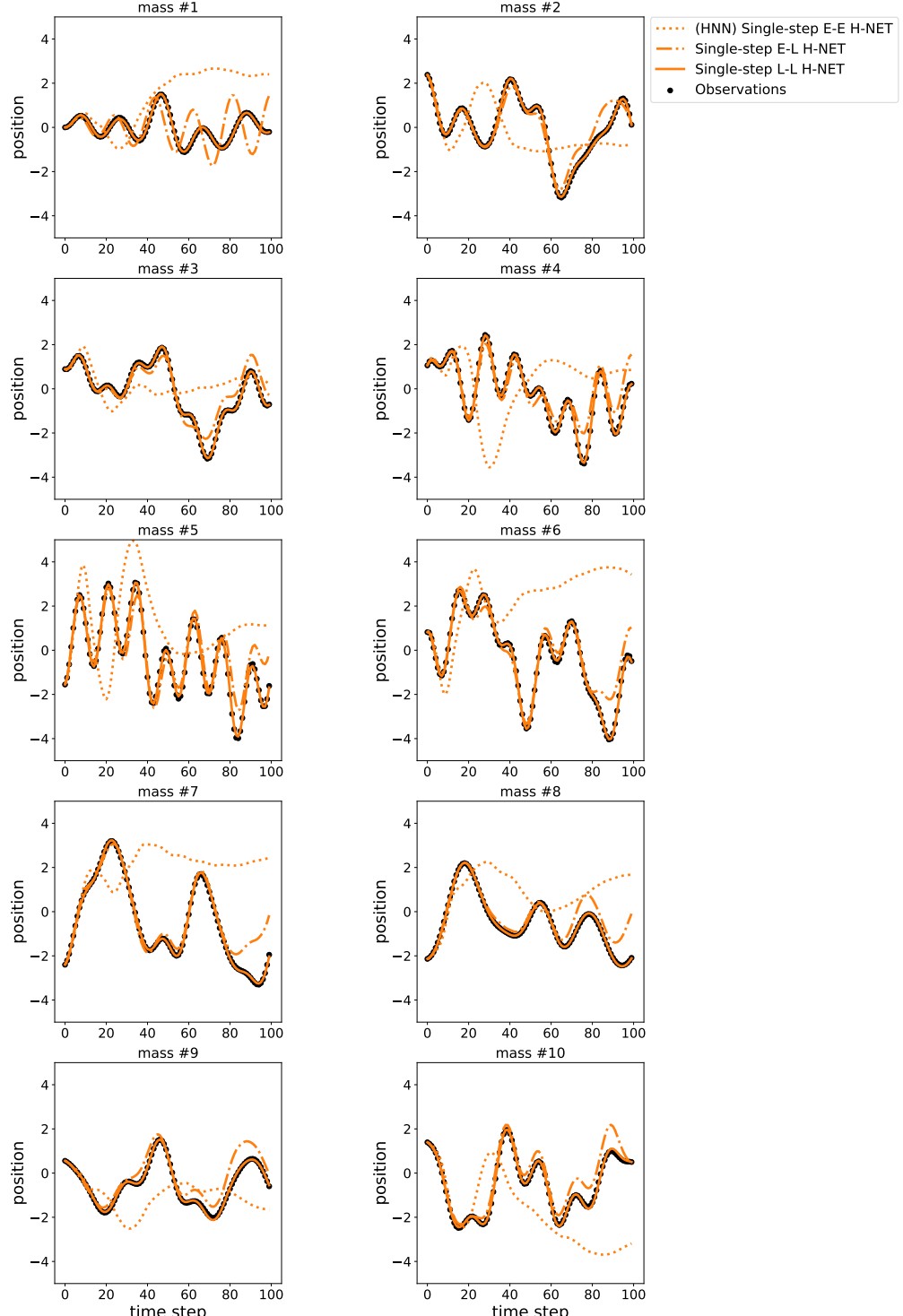

Figure 5: Extension of Figure 1 to 10 masses on the chain (1st being the closest to one end, and 10th being in the center).

## D.2 Noisy data (section 4.2)

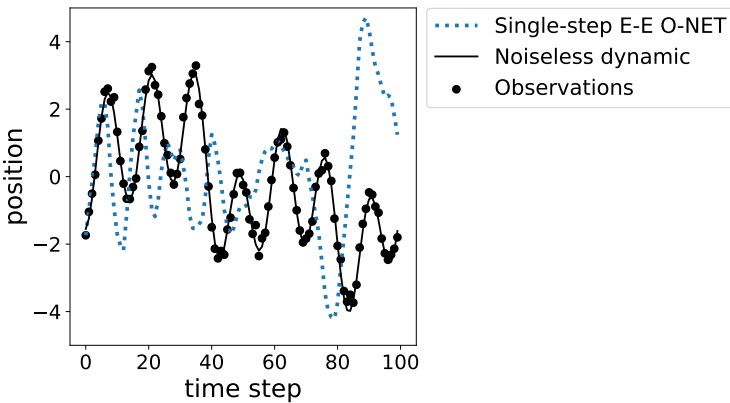

Figure 6: Single-step E-E O-NET

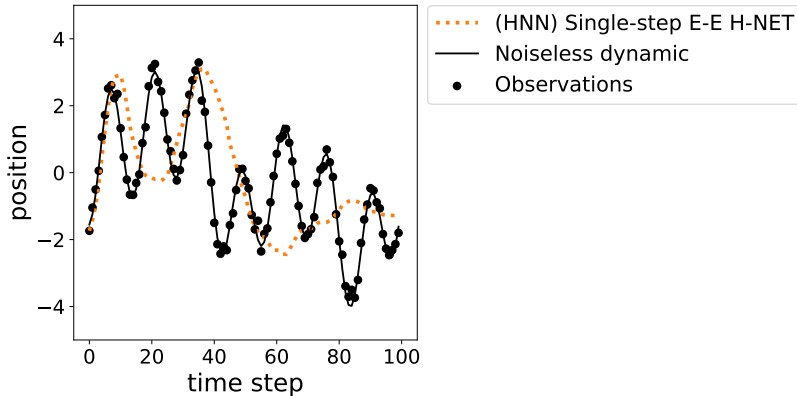

Figure 7: Single-step E-E H-NET

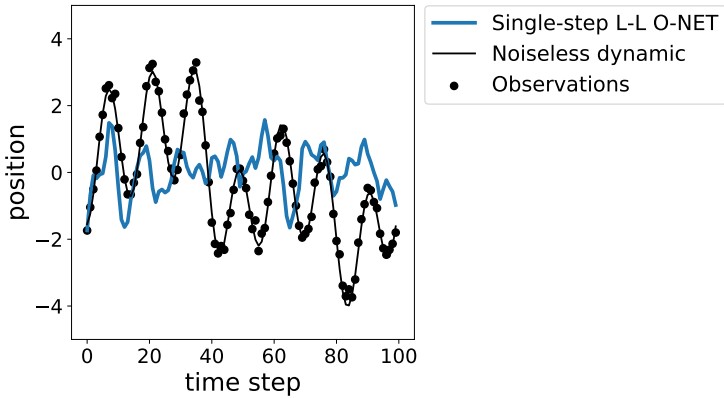

Figure 8: Single-step L-L O-NET

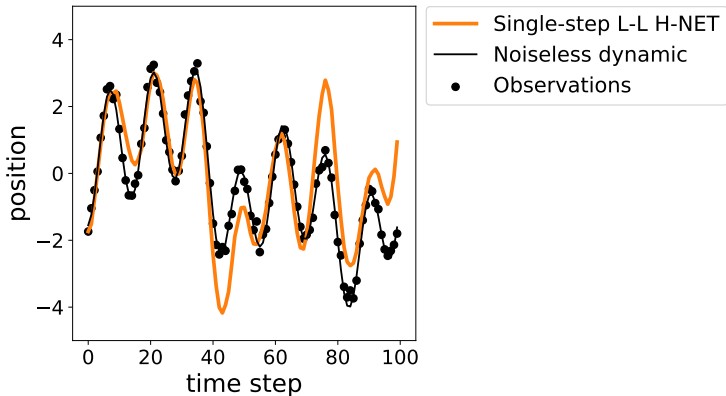

Figure 9: Single-step L-L H-NET

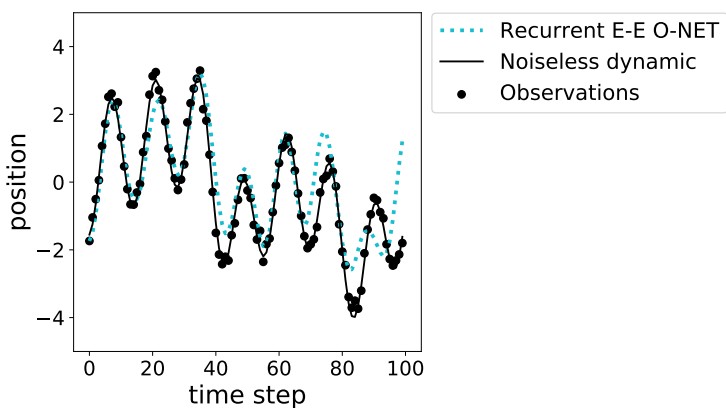

Figure 10: Recurrent E-E O-NET

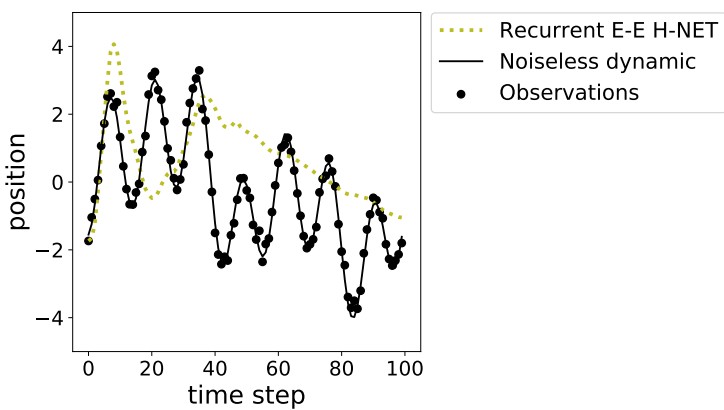

Figure 11: Recurrent E-E H-NET

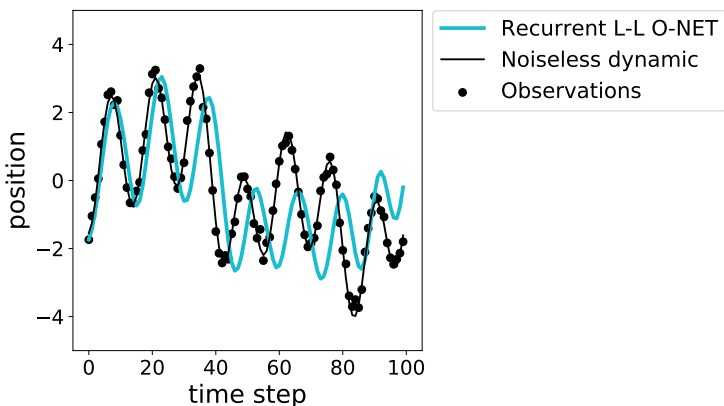

Figure 12: Recurrent L-L O-NET

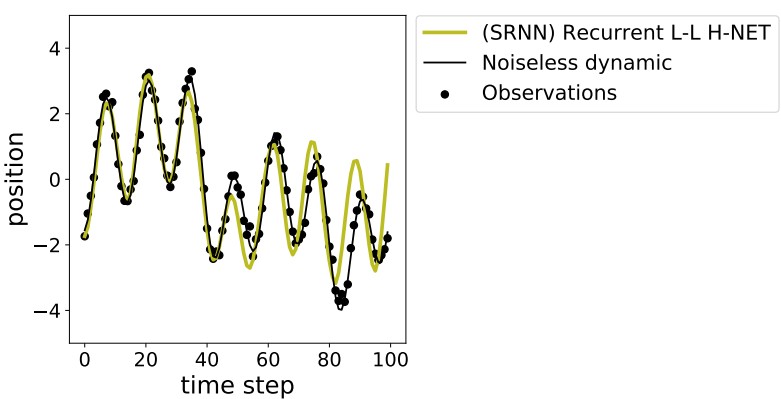

Figure 13: (SRNN) Recurrent L-L H-NET

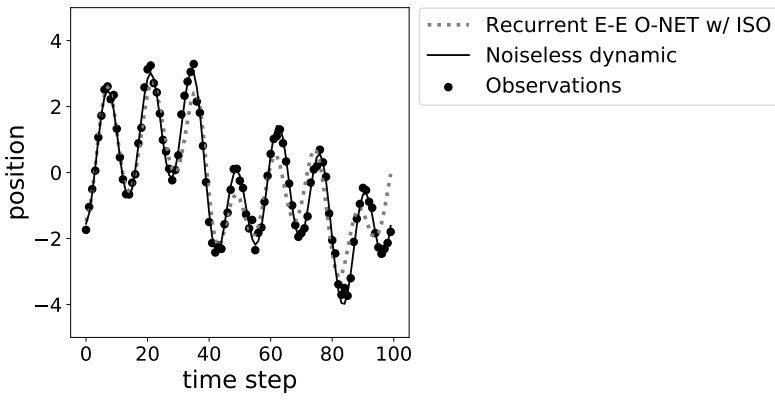

Figure 14: Recurrent E-E O-NET w/ ISO

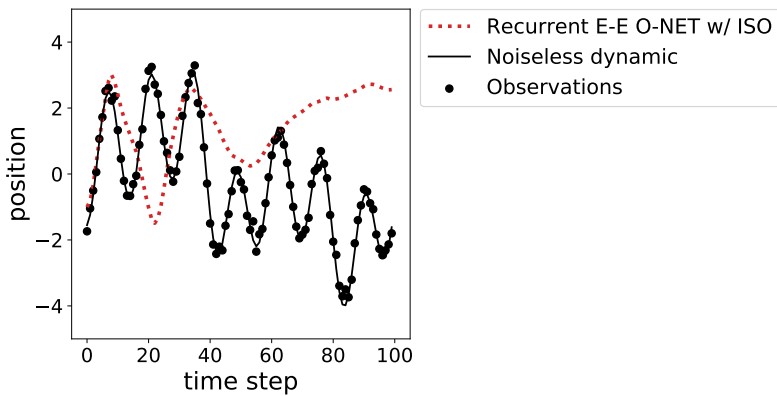

Figure 15: Recurrent E-E H-NET w/ ISO

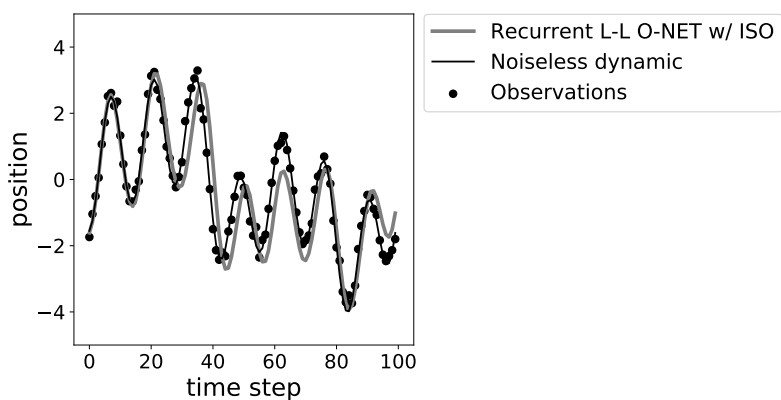

Figure 16: Recurrent L-L O-NET w/ ISO

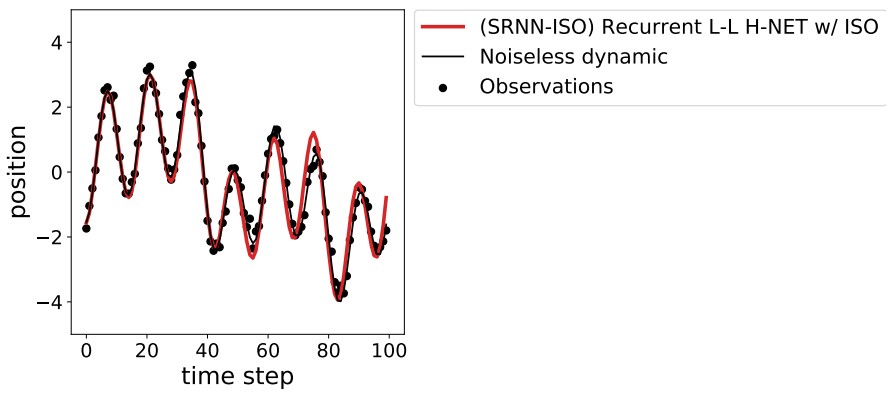

Figure 17: (SRNN-ISO) Recurrent L-L H-NET w/ ISO

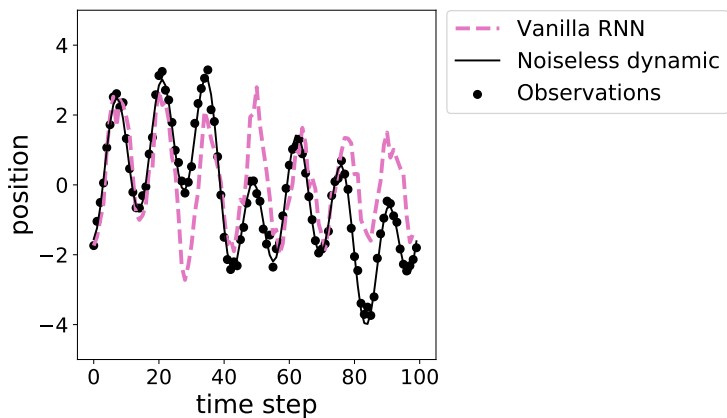

Figure 18: Vanilla RNN

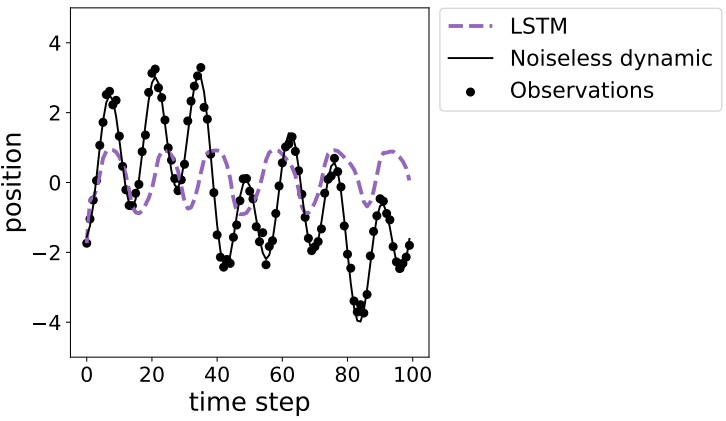

Figure 19: LSTM

# E ADDITIONAL PLOTS OF THE THREE-BODY EXPERIMENTS

In each of the plots below, the three dashdot curves represent the ground truth trajectories of the three masses, and the three sequences of dots are the predictions made by each method.

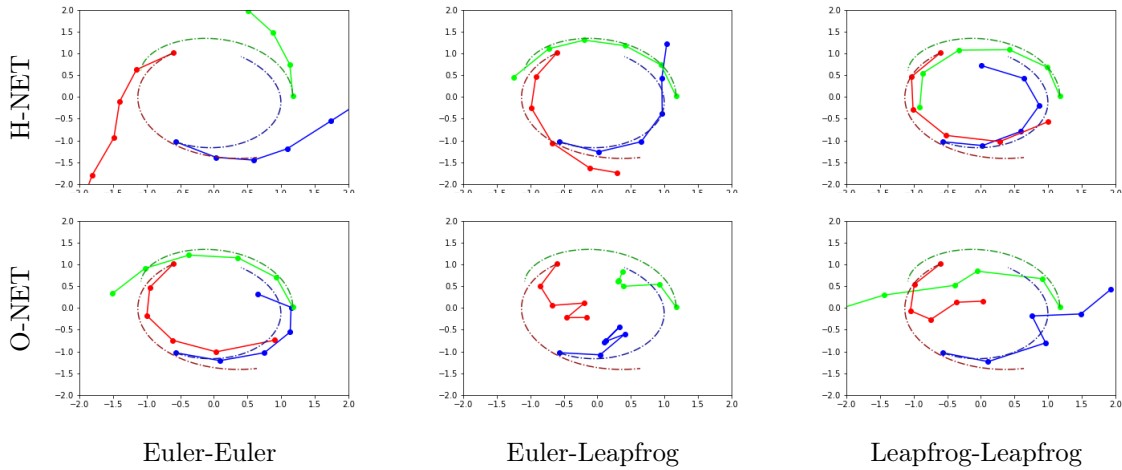

Figure 20: Actual versus predicted trajectories of the three-body system by the various single-step-trained methods.

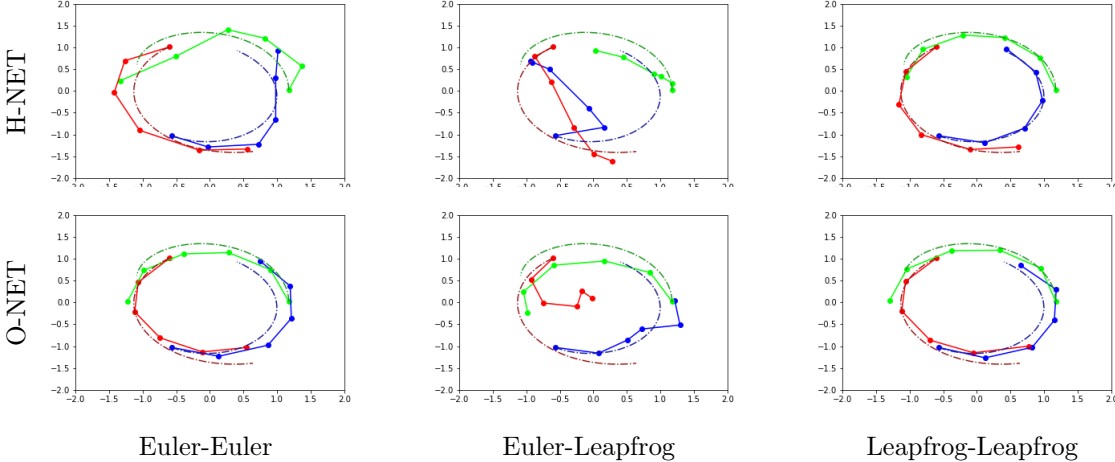

Figure 21: Actual versus predicted trajectories of the three-body system by the various recurrently trained methods.

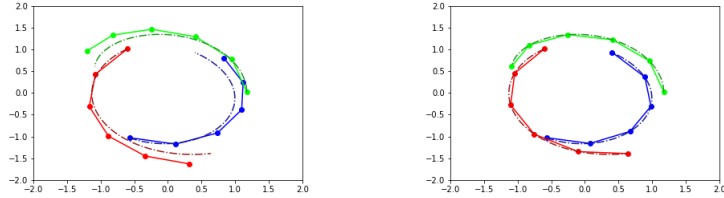

Figure 22: Actual trajectory versus the trajectory simulated by the leapfrog integrator with time-step 1 (left) and 0.1 (right).

# F    ADDITIONAL PLOTS OF THE HEAVY BILLIARD EXPERIMENT

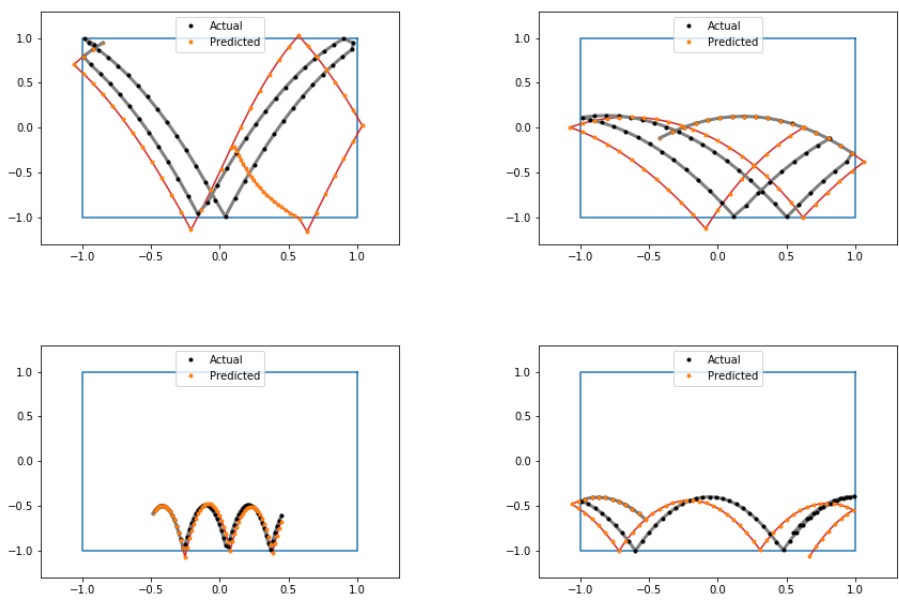

Figure 23: SRNN with a rebound module that does not learn $\alpha$ (and effectively treats $\alpha = 1$).

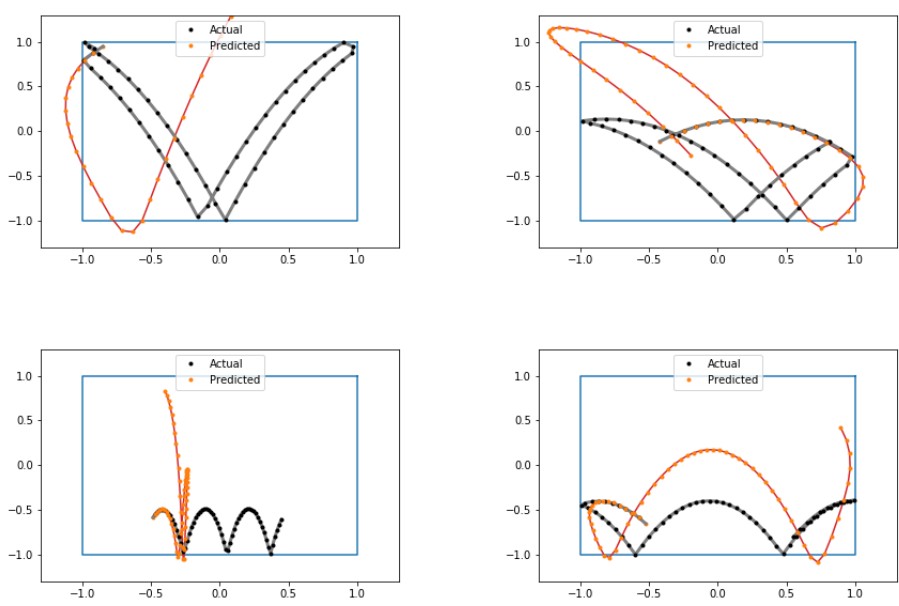

Figure 24: SRNN without the rebound module.

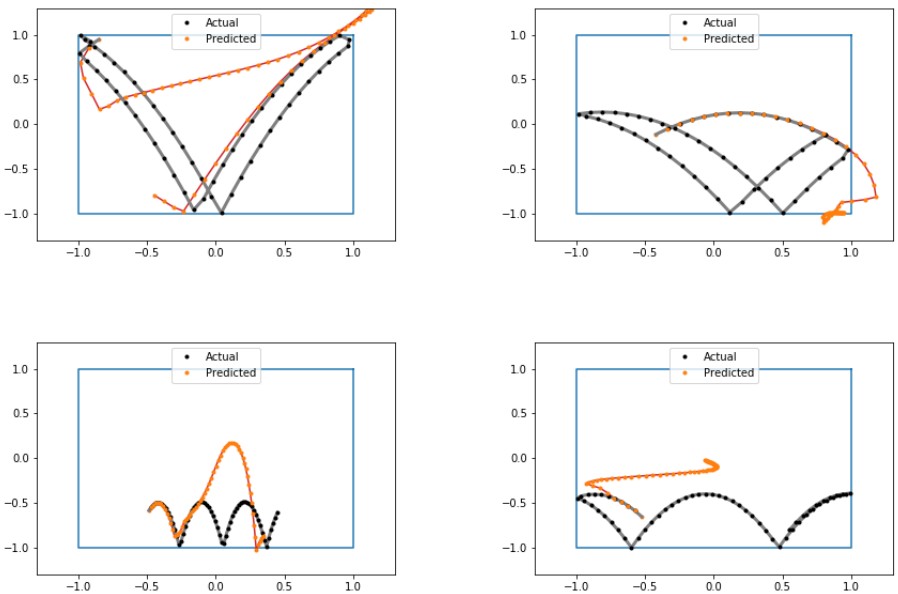

Figure 25: Recurrent L-L O-NET with the rebound module.

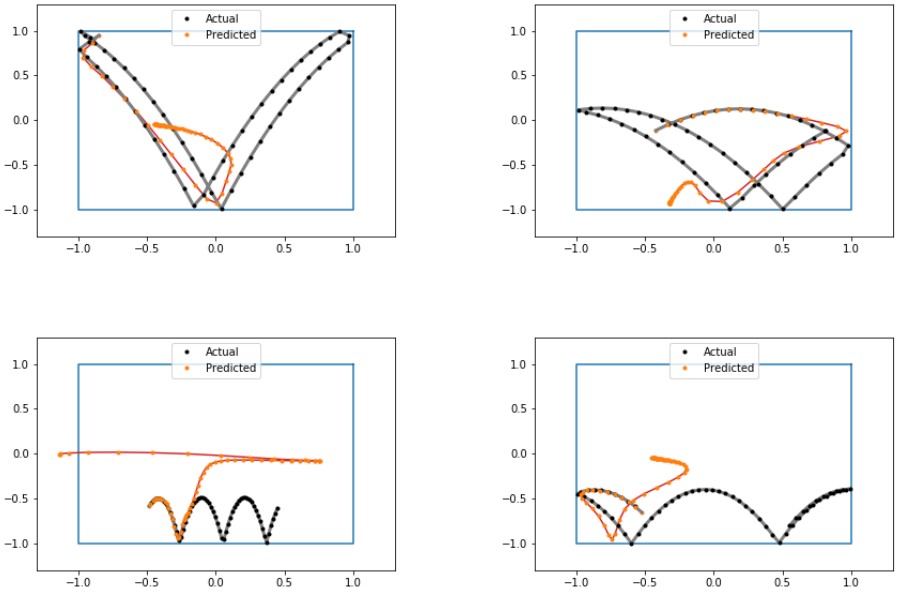

Figure 26: Vanilla RNN.

