# OpenReview forum: "Symplectic Recurrent Neural Networks"
_ICLR.cc/2020/Conference — Accept (Spotlight)_

### Official Review · AnonReviewer1 · 2019-10-23
**Official Blind Review #1**

**Rating:** 6

**Review:**

This paper introduces SRNN to model the hamiltonian of a dynamical system. Authors break down the design choices made in the algorithm and validate each one through experiments.

I like the motivation of the paper of solving general physics problems using neural networks. The paper is well-written and the ideas are communicated properly.

However, I have some concerns regarding the experimental results:

In Figure 1, even though the fifth mass seems to follow the exact pattern of observations for single-step L-L H-NET, the L2 error is increasing after each time-step suggesting that the fifth mass might not be the best one to consider for comparison between different algorithms here. Would be nice if a comparison between all trajectories was presented and discussed (perhaps in the Appendix)

The idea tested in section 4.2 seems not novel since previous works have already used recurrence to mitigate problems faced using one-step training. (https://arxiv.org/abs/1902.09689)

I like the optimization method proposed in Section 4.3 and the results in Table 1 and 2 seems to justify its effect. Does this approach also handle degenerate cases in which different initial states can lead to the same trajectories after some amount of time? To me, it seems like a boundary on the noise variance should be assumed. Otherwise, no amount of optimization would be actually able to retrieve the correct initial state. Is this true or am I missing something?



-minor comments:

1. The conclusion section is missing from the paper. It would be nice to recap your findings there.
2.  In section 2, the "universality property" of Niu's recurrent model should be explained or referenced.
3. In Figure 1, the second "Left:" should be changed to "Right:".
4. In Table 2, the Error std. for O-NET (E-L) should be bold not H-NET (L-L) unless it was intended to highlight the values for the model with the lowest mean error. (please clarify this in the paper.)
5. Section 6: "We focus on this section" should be changed to "We focus in this section".


Overall I think the work is interesting but it lacks some justifications regarding the claims made (as mentioned above), and although it is generalizable to other tasks and systems, it does not have sufficient novelty in its algorithm and approach.

As of now, I am recommending a rejection, but I am willing to reconsider my score should the authors address the above concerns.


----------------------------------------------------------------------------------------------------------------------------
Updates:

Thank you for your response and addressing my concerns in the revised version of the paper. I also see new updates to the text which has improved the readability considerably. Thank you for your work.
I have updated my score accordingly.




**Experience Assessment:**

I have read many papers in this area.

**Review Assessment: Checking Correctness Of Derivations And Theory:**

I assessed the sensibility of the derivations and theory.

**Review Assessment: Checking Correctness Of Experiments:**

I assessed the sensibility of the experiments.

**Review Assessment: Thoroughness In Paper Reading:**

I read the paper thoroughly.

---

> ### Author Response · Authors · 2019-11-09
> **Response to Reviewer #1**
>
> Thank you for the detailed comments!
>
> 1) On the consistency between the “fifth-mass” plot and the L2 error plot in Figure 1:
> In Figure 1 (right), indeed, single-step L-L H-NET predicts almost exactly the trajectory of the fifth mass on the chain in the first 100 time steps. This is consistent with Figure 1 (left), which plots the L2 / Root Mean Squared error over the positions of all 20 masses, and shows that the single-step L-L H-NET accumulates error much more slowly over time than the two other models. As you pointed out correctly, its prediction error still grows over time, which is not quite visible in Figure 1 (right), although we can see its deviation from the actual trajectory if we plot them for longer time. By the way, when noise is added, the “fifth mass plots” in Appendix D indicate that single-step L-L H-NET performs visibly worse, especially compared to the recurrently trained models. In fact, the fifth mass' position turns out to be a relatively good indicator of the prediction quality. They work better than the masses closer to the ends, for example, whose trajectories are easier to predict. But we appreciate your suggestion, and yes, we will add plots in the appendix for the trajectories of other masses on the chain.
>
> 2) On the relation to Antisymmetric RNN & one-step v.s. multi-step / recurrent training:
> Thanks for mentioning the paper of Antisymmetric RNN, and we actually discussed it a bit in section 2. This model was not developed for the task of learning and predicting dynamics from data, but indeed it would be interesting to test it experimentally. By the way, in the field of numerical ordinary differential equations, there is also the distinction between one-step and multi-step methods, but in that context they have different meanings: they refer to whether to compute the current step based on the function evaluation of only the previous step, or a few previous steps. In fact, both Euler’s integrator and the leapfrog integrator belong to the family of one-step integrators, but nonetheless we can perform multi-step / recurrent training with them, according to our paper’s terminology. Our first contribution is then to demonstrate the advantage of multi-step over one-step training to combat observation noise in the data. Moreover, we find that for learning noisy Hamiltonian dynamics, multi-step training is compatible with initial state optimization, which is somewhat remarkable given prior work in the parameter estimation literature on the difficulties of initial state optimization in other setups, as we discuss in Appendix B.  We then give a heuristic argument in Appendix C for why initial state optimization may be easy in learning Hamiltonian dynamics – in short, we think that symplecticness could render the optimization problem close to convex.
>
> 3) On the effect of noise on initial state optimization & degenerate cases:
> It is a very good point that noise level affects how successful the initial state optimization is. In Appendix C, we actually show that how close to convex the initial state optimization problem is depends on the amount of noise. Hence we agree that if noise level is too high, the problem can certainly become hard or impossible. The question about degenerate cases is interesting too. But we want to mention that our goal is not necessarily to find the correct initial state, but to find one that allows us to accurately predict the rest of the trajectory. If the system is such that different initial states converge to the same trajectories after a while, then we do not need to recover the exact initial state in order to predict the future trajectory. This can happen in dissipative systems, for example.
>
> 4) A highlight of the contribution to handling stiff Hamiltonian systems:
> Finally, we just want to add that besides proposing SRNN and demonstrating the importance of symplectic integration, multi-step training and initial state optimization in learning the dynamics of noisy and complex Hamiltonian systems, another novel contribution is to propose an augmentation of SRNN that can successfully learn perfect rebound. Perfect rebound is a prototypical example of stiffness in ODE systems, which are notoriously troublesome for numerical integrators of differential equations and therefore one would naturally expect them to challenge learning algorithms based on differential equations fundamentally.
>
> We appreciate your comments and suggestions very much, and will make changes to the typos and the other errors that you pointed out.

---

> ### Author Response · Authors · 2019-11-13
> **A revision has been uploaded**
>
> Dear reviewer,
>
> We have uploaded a revised version of our paper, in which we made edits to reflect the issues you pointed out. Thanks again for your helpful suggestions!
>
> In particular, we added plots in Appendix D.1 that extend the "fifth-mass plot" in Figure 1 (right) to ten masses on the chain (with the 1st being on one end, and the 10th being in the center, as there are twenty masses in total). It turns out that the contrast among the three methods is consistent for all these masses, with L-L H-NET being the best (almost perfect), and E-L H-NET being the second best. This is also consistent with Figure 1 (left), which shows that L-L H-NET achieves much smaller (though still slowly growing with time) error than the other two methods. This supports our message that symplectic integration is crucial for learning the Hamiltonian.
>
> Hope our comments help, and we would love to discuss further if you have any other questions or thoughts about our work. Thanks!

---

### Official Review · AnonReviewer2 · 2019-10-23
**Official Blind Review #2**

**Rating:** 8

**Review:**

This paper proposes to represent a Hamiltonian model of a physical system by a neural network. The parameters are then adjusted, so that the observations are considered maximally likely under a probabilistic model. The novelty is to consider a symplectic Leapfrog integration scheme for the Hamiltonian system, which is known to conserve important quantities such as volume in the state space. The proposed approach is shown to outperform the recent work "Hamiltonian Neural Networks" by a large margin on mass-spring chain dynamics and three body systems. The approach can even handle stiff dynamical systems such as bouncing billiards.

Overall, the work is solid contribution and a reasonable improvement over the recent work on HNNs is demonstrated. Therefore I recommend acceptance of the paper. However, I have some fundamental doubts on the motivation on this line of works. This might be, because I'm not too familiar with the subject, and I'm willing to increase my score if the doubts are cleared.

In the shown examples, to best of my knowledge, the "exact" Hamiltonians describing the physics of the system are well-known. Therefore, I'm unsure what is the advantage of trying to learn physical laws, that are already well understood. The paper argues that the learned Hamiltonian will correct for errors in the discretization, but one could instead use a better integration scheme or a finer time-discretization, based on classical theory which has been developed over the last 50 years which comes with strong convergence guarantees and error bounds. I would have liked to see a stronger motivation, why it is interesting to learn an Hamiltonian of a system, where the exact Hamiltonian is already known. It would also be enlightening to see some plots, which illustrate how "far" the learned Hamiltonian is from the analytical one.

Of course, one might argue that the ultimate goal is to have a learning based approach discover physical laws so far unknown to humans,  just from observations. But it is unclear why the inductive bias that the observations are generated by a Hamiltonian might be reasonable. It could very well be, that the law cannot be described by a Hamiltonian system.

From a high-level point of view, one might even argue that it is not too surprising that one can fit a parametrized Hamiltonian to observations generated by a Hamiltonian system better than a general purpose function approximator without such an inductive bias or better than a system based on a naive/unsuitable non-conservative integrator.

As a remark, often the exact Hamiltonian is known to be (strictly) convex. I'm wondering whether convex function approximators such as convex neural networks could provide an even stronger inductive bias. But it might be that a general purpose RNN can account better for the discretization errors.

**Experience Assessment:**

I do not know much about this area.

**Review Assessment: Checking Correctness Of Derivations And Theory:**

I assessed the sensibility of the derivations and theory.

**Review Assessment: Checking Correctness Of Experiments:**

I assessed the sensibility of the experiments.

**Review Assessment: Thoroughness In Paper Reading:**

I made a quick assessment of this paper.

---

> ### Author Response · Authors · 2019-11-09
> **Response to Reviewer #2**
>
> Thanks for your comments! Below are some of our thoughts.
>
> 1) On whether we want to learn known or unknown Hamiltonians:
> We appreciate that you pointed out the two different understandings of the value of our work. Indeed, we are mainly motivated by learning and predicting the dynamics of systems with unknown Hamiltonians, and the ability of our models to compensate for numerical discretization errors is more like an interesting side finding, which could be of independent interest to people applying machine learning to solving differential equations numerically. We focus on simulated systems with known dynamics because even for these systems previous methods do not perform very well. We demonstrate that an important reason for their failure is a lack of attention to the numerical properties of Hamiltonian systems, and show that it is fixable. In addition, we show that multi-step training and initial state optimization are crucial when observation noise is present in the data. Moreover, we show that our model, SRNN, can be augmented to learn perfect rebound, which is a prototypical example of stiffness in Hamiltonian systems and is expected to be challenging to learning algorithms based on ODEs. We consider all of these as substantial steps towards addressing more realistic problems.
>
> 2) On the Hamiltonian assumption as an inductive bias:
> Modern physics suggests that nearly all physical phenomena (classical, quantum, or relativistic) admit Hamiltonian formulations when one considers the right set of variables. It probably takes more than an SRNN to learn all of them well, but we think SRNN is a decent generic model that can serve as a basis for more specialized ones. The appearance of non-Hamiltonian dynamics can arise when one uses an incorrect (non-canonical) or incomplete set of variables, or when one ignores how the system exchanges conserved quantities with its environment. For instance, a computer can simulate a non-Hamiltonian dynamics despite the fact that it does so by harnessing physical processes that follow a Hamiltonian dynamics.  According to this viewpoint, Hamiltonian dynamics is an attractive inductive bias because we know that, at some level, reality follows Hamiltonian dynamics. The difficulty is to infer the correct variables, something we plan to attack in a forthcoming paper. The present paper only addresses a first step, that is, successfully model and learn a Hamiltonian dynamical system with known variables.
>
> 3) On the convexity of Hamiltonian functions:
> We don’t think that Hamiltonian functions are necessarily convex. For example, a particle moving in a non-convex potential (e.g. originating from gravity or Coulomb force, which obey the inverse-square law) or a system of interacting particles can have non-convex Hamiltonian functions. However, it is a good point that if we have extra knowledge of the Hamiltonian function, such as having a kinetic component that is quadratic in the momentum variables, then it is reasonable to try building it into the neural network with which we approximate the Hamiltonian function.

---

### Official Review · AnonReviewer3 · 2019-11-04
**Official Blind Review #3**

**Rating:** 8

**Review:**

Summary:

The paper improves upon Hamiltonian Neural Networks to model physical systems from observed trajectories. Specifically, the authors propose to use (i) better integrator, (ii) multi-step learning, and (iii) initial state optimization. Authors experimentally show that all these improvements are beneficial in modelling complex and noisy Hamiltonian systems.


My Comments:

I come from RNNs background and I am not an expert on physical systems. But the paper is extremely well written that I can easily follow. Experiments are convincing.

I do not have any issues with the claims.

Are the authors willing to release their code to reproduce the results?

========================

Post rebuttal: I stand by my decision.


**Experience Assessment:**

I do not know much about this area.

**Review Assessment: Checking Correctness Of Derivations And Theory:**

I assessed the sensibility of the derivations and theory.

**Review Assessment: Checking Correctness Of Experiments:**

I assessed the sensibility of the experiments.

**Review Assessment: Thoroughness In Paper Reading:**

I read the paper at least twice and used my best judgement in assessing the paper.

---

> ### Author Response · Authors · 2019-11-09
> **Response to Reviewer #1**
>
> Thanks for the compliments : )
>
> Yes, our code is on GitHub and we will publicize it after de-anonymization.

---

### Author Response · Authors · 2020-04-29
**GitHub repository**

The code is available at https://github.com/zhengdao-chen/SRNN.git.

---

### Decision · Program_Chairs · 2019-12-19

**Decision:**

Accept (Spotlight)

**Comment:**

This paper proposes a novel architecture for learning Hamiltonian dynamics from data. The model outperforms the existing state of the art Hamiltonian Neural Networks on challenging physical datasets. It also goes further by proposing a way to deal with observation noise and a way to model stiff dynamical systems, like bouncing balls. The paper is well written, the model works well and the experimental evaluation is solid. All reviewers agree that this is an excellent contribution to the field, hence I am happy to recommend acceptance as an oral.